# β-hydroxybutyrate Impedes the Progression of Alzheimer’s Disease and Atherosclerosis in ApoE-Deficient Mice

**DOI:** 10.3390/nu12020471

**Published:** 2020-02-13

**Authors:** Manigandan Krishnan, Jong Su Hwang, Mikyung Kim, Yun Jin Kim, Ji Hae Seo, Jeeyoun Jung, Eunyoung Ha

**Affiliations:** 1Department of Biochemistry, School of Medicine, Keimyung University, Daegu 42601, Korea; biomani1@gmail.com (M.K.); z_zone8863@naver.com (J.S.H.); salang0620@nate.com (Y.J.K.); 2Clinical Medicine Division, Korea Institute of Oriental Medicine, Daejeon 34054, Korea

**Keywords:** Alzheimer’s disease, β-hydroxybutyrate, atherosclerosis, choroid plexus, apolipoprotein-E

## Abstract

β-hydroxybutyrate (β-OHB) has been shown to exert an anti-inflammatory activity. Apolipoprotein-E (ApoE) is strongly associated with atherosclerosis and Alzheimer’s disease (AD). This study aimed to explore the therapeutic effect of β-OHB in the brain and the aorta of high-fat diet (HFD)-fed ApoE-deficient mice. We found in Apo-E deficient mice that β-OHB attenuated lipid deposition in the choroid plexus (ChP) and decreased amyloid plaque in the substantia nigra pars compacta. We also found decreased CD68-positive macroglia infiltration of the ChP in β-OHB-treated ApoE-deficient mice. β-OHB treatment ameliorated IgG extravasation into the hippocampal region of the brain. In vitro study using ChP mice cell line revealed that β-OHB attenuated oxidized low-density lipoprotein-induced ApoE-specific differentially expressed inflammatory ChP genes. Treatment with β-OHB reduced aortic plaque formation without affecting blood lipid profiles and decreased serum production of resistin, a well-established risk factor for both AD and atherosclerosis. Thus, the current study suggests and describes the therapeutic potential of β-OHB for the treatment of AD and atherosclerosis.

## 1. Introduction

World Alzheimer Report 2018 states that 50 million people worldwide are living with dementia, and this number will more than triple to 152 million by 2050. The total estimated worldwide cost of dementia in 2018 is one trillion US dollars, which is expected to rise to two trillion US dollars by 2030. Although the understanding of the neurobiology and pathogenesis of Alzheimer’s disease (AD), the most common form of dementia, has recently been greatly increased, no curative, but only a few symptomatic treatments, are currently available. Thus, parallel to the effort towards the development of drugs for the treatment of AD, effort in developing preventive therapeutic strategies, such as nutritional therapies and caloric restriction, have been extensively investigated and implemented [1].

Human apolipoprotein-E (ApoE) is a polymorphic multifunctional protein with isoforms of ApoE2, ApoE3, and ApoE4, and is strongly implicated in two major inflammatory diseases, AD and atherosclerosis. People with apoE4 allele are susceptible to late onset AD [2,3]. Atherosclerosis, characterized by lipid plaque formation in the aorta, is the leading cause of cardiovascular disease [4]. Action mechanisms of ApoE include involvement of lipoprotein receptors, effects on cholesterol efflux, and maintenance of the blood-brain barrier [5,6,7].

β-hydroxybutyrate (β-OHB) is a major component of ketone bodies. The efficacy of a ketogenic diet in treating childhood refractory epilepsy and mild cognitive impairment that usually precedes the onset of AD has been well established, and paved the way to evaluate the potential therapeutic effect of β-OHB [8,9,10,11]. However, a common mode of action of β-OHB in AD and other neurodegenerative diseases, as well as atherosclerosis, has yet to be fully elucidated.

Our data in the current study indicate that β-OHB delays the progression of AD by inhibiting lipid accumulation and inflammatory reactions in the brain, and of atherosclerosis by decreasing plaque formation in the aorta.

## 2. Materials and Methods

### 2.1. Materials and Reagents

β-hydroxybutyrate (β-OHB), Oil Red O (ORO), Congo red, and other chemicals were purchased from Sigma Aldrich (St. Louis, MO, USA) unless otherwise specified. Anti-CD68 antibody (ab125212) was purchased from Abcam (Cambridge, UK). Normal mouse IgG (sc-2025) antibody was purchased from Santa Cruz Biotechnology (Dallas, TX, USA).

### 2.2. Animals 

All experimental protocols and procedures were approved by and performed according to the guidelines of the experimental Animal Care Committee at Keimyung University, School of Medicine (KM-2018-09). Six-week-old male ApoE^−/−^ (C57BL/6J background) and C57BL/6J mice were purchased from the Jackson laboratory (Sacramento, CA, USA). Two weeks after acclimation, C57BL/6J mice were fed normal chow diet (NCD, *n* = 10) and ApoE^−/−^ high-fat diet (HFD, *n* = 20) for 8 weeks. Eight weeks later, mice in the HFD group were randomly divided into two groups (*n* = 10 for each group), one with phosphate buffered saline (PBS)-containing Alzet® osmotic minipumps (#1004, DURECT Corp., Cupertino, CA, USA), and the other with β-OHB (1.5 mmol/kg/day in PBS)-containing Alzet® osmotic minipumps implanted subcutaneously onto the left flank. Four weeks after first implantation, Alzet® were removed and newly prepared second Alzet® were implanted subcutaneously onto the right flank. Mice were sacrificed four weeks after second implantation following a 12 h overnight fasting. Serum was collected by centrifugation at 2000 g for 10 min. Aortic and whole brain tissues were fixed and processed for histopathological analyses, and remaining tissues were quick frozen with liquid nitrogen and stored at −80 °C until further analysis.

### 2.3. Cell Culture

ECPC4 cells (mouse choroid plexus cell line) were purchased from RIKEN cell bank (Tsukuba, Japan) and cultured in RPMI 1640 supplemented with 10% fetal bovine serum, 100 units/mL penicillin, and 100 mg/mL streptomycin. Cells were maintained at 37 °C in a humidified atmosphere with 5% CO_2_. 

### 2.4. Quantitative Real-Time PCR 

Total RNA was isolated from the ECPC-4 cells with Trizol reagent (Invitrogen). Reverse transcription was performed to yield cDNA using a GoTaq® PCR Core System (Promega, Korea). The RNA (1 mg) was performed using SYBR green reagents (QPK-201, Toyobo, Japan) on a LightCycler^®^ 96 Instrument (Roche Diagnostics Corporation, Indianapolis, IN, USA). The β-actin was used as an internal control. Data were analyzed using Light Cycler 96 Instrument software. Relative gene expression was calculated using the 2 ^−ΔΔCt^ method [12]. 

### 2.5. Immunohistochemistry

Congo red staining for Aβ aggregation was performed according to the established protocol [13]. Briefly, slides were removed from −80 °C storage, fixed with 4% paraformaldehyde for 10 min, and washed in tap water. The sections were then stained with Congo red (1% w/v), differentiated with alkaline alcohol (1% potassium hydroxide in 80% ethanol), and then counterstained with hematoxylin for 4 min before they were dehydrated and mounted. Plaques were observed under a light microscope and the total area (%) of Aβ plaques per section or per brain area were quantified using ImageJ. CD68 (diluted 1:500), normal mouse IgG (diluted 1:200) expressions in the mouse coronal sections were determined by immunohistochemistry according to the previously described method [14]. The cryo-sectioned slides were fixed with 4% paraformaldehyde, quenched with 0.3% H_2_O_2_ for 5 min, washed with PBS 3 times, blocked with BSA (3% w/v in PBS), and then incubated overnight with an anti-CD68 primary antibody (1:1000). Subsequently, the stained sections were incubated with HRP conjugated secondary antibody (1:200) for 1h and the immunocomplexes were visualized by 3,3-diaminobenzidine (DAB) substrate, and all sections were counterstained with hematoxylin prior to mounting. The positive signals were visualized by light microscope (Leica, CA, USA) and the stained area was quantified by ImageJ software.

### 2.6. Immunofluoresence and Thioflavin-S Staining

Frozen sections were incubated in 0.3% Triton X-100 for 5 min and blocked by incubation in blocking solution (3% bovine serum albumin, 0.05% Tween-20 in TBS) for 60 min. Sections were then incubated overnight with AT8 (phosphor-tau at Ser ^202^/Thr ^205^, Thermo Fisher Scientific, Rockford, IL, USA) at 4 °C. On the second day, sections were washed in 10 mM PBS, followed by incubation with Alexa-568 conjugated goat anti- mouse IgG (diluted 1:500, Thermo Fisher Scientific) in blocking solution for 1 h at room temperature. Sections were subsequently washed, for double labeling with AT8, and stained with 0.3% thioflavin-S (Sigma-Aldrich) for 5 min. Stained sections were washed in 80% ethanol twice for 3 min, mounted with aqueous mounting medium (Vector Laboratories, Burlingame, CA, USA, H-5501), and then captured with confocal microscopy.

### 2.7. Oil Red O (ORO) Staining 

Atherosclerotic lesions deposition in the aortic root was analyzed as described previously [15]. In brief, the fixed aorta was opened longitudinally from the arch to the iliac bifurcation to obtain a flat preparation. The flattened aortic tissues were then stained with ORO (0.5% w/v in isobutanol) for 20 min. After washing, the stained lesion area was examined using a light microscope (Leica, CA, USA). The aortic root sections were stained with hematoxylin and eosin (H&E). Similarly, mouse brain tissues were embedded and frozen in optimal cutting temperature (O.C.T.) compound at −80 °C. Twenty micrometer whole mouse brain coronal sections were prepared using Cryotome (MEV, SLEE, Mainz) and stained with ORO. The total area of lipid deposits at aorta and ChPs regions were quantified using ImageJ (National Institute of Health(NIH), Bethesda, MD, USA) and expressed as the plaque area percentage. 

### 2.8. Low Density Lipoprotein (LDL) Isolation and Oxidation

Blood was drawn from healthy voluntary human subjects in the fasting state. Blood samples were collected into sterile ethylenediaminetetraacetic acid (EDTA) containing tubes, and plasma was separated for 20 min at 2,000 g centrifugation. LDL was isolated by sequential unltracentrifugation, yielding LDL at a final density of 1.019 to 1.063 g/mL with potassium bromide in EDTA-saline. The isolated LDL was dialyzed 1 mM EDTA buffer (pH 8.0) and then oxidized with CuSO_4_ (Sigma-Aldrich, USA). LDL oxidation with copper was performed by incubation of LDL (0.2 mg of LDL protein/mL) with 5 μM CuSO4 in PBS (pH 7.4) for 4 hat 37 °C. The oxidized LDL was dialyzed with 0.15 M NaCl solution containing 0.01% EDTA buffer (pH 7.0) for 36 h at 4 °C.

### 2.9. Lipid Profiles

Blood samples were collected into tubes and centrifuged at 3000 rpm for 10 min. The levels of total cholesterol, LDL-cholesterol and HDL-cholesterol in serum were measured using kits purchased from BioVision Ltd (San Francisco, CA, USA) according to the manufacturer’s instructions. Triglyceride level was measured with a Triglyceride Assay Kit (Cayman Chemical, Ann arbor, MI, USA). All assays were performed according to the manufacturer’s instructions. The absorbance values were determined with an ELISA microplate reader (Biochrom, Cambridge, UK).

### 2.10. Analysis of Leptin and Resistin Levels

The levels of leptin and resistin in plasma were analyzed by using the Bio-Plex Pro Mouse Diabetes Set immunoassay kit (Bio-Rad Laboratories, Hercules, CA, USA) according to the manufacturer’s instructions.

### 2.11. Statistical Analysis

All data were exported to GraphPad Prism v8.0 (GraphPad Software) for statistical analyses. Values represent the mean ± standard deviation (SD). Statistical significance was determined based on *p*-values obtained from one-way ANOVA with Tukey’s test.

## 3. Results

### 3.1. β-OHB Attenuated HFD-Induced Lipid Deposition, Amyloid Plaque Formation in the ChP, and Tau Accumulation in the Hippocampal Region of ApoE^−/−^ Mice

The overall in vivo experimental procedure is illustrated in Figure 1A. Given the pathophysiological role of ApoE on both atherosclerosis and AD, we first focused our view particularly on the brain’s ChP region, the blood-cerebrospinal fluid (CSF) barrier, of ApoE^−/−^ mice to elucidate the neuroprotective effect of β-OHB. As expected, HFD increased lipid deposition in the ChP region of ApoE^-/-,^ while treatment of β-OHB attenuated HFD-induced lipid deposition (Figure 1B,C). We next employed Congo red staining to determine the presence of AD-plaques such as amyloid-beta (Aβ) accumulation, which also showed decreased plaque formation in the substantia nigra pars compacta (SNR) region of ApoE^−/−^ mice when compared with control ApoE^−/−^ mice (Figure 1B). It is well documented that the degree of ChP lipid deposit-associated chronic inflammation correlated with all AD-related neuropathology. Overwhelmed lipid deposits alter the immunological interface in CSF, and we noticed an increase in the expressions of CD68 macrophage in ChP of ApoE^−/−^ mice (Figure 1C,D), representing that lipid deposits might be colocalized with the macrophages to induce complex pathological signaling in brain CSF. Notably, β-OHB treatment reduced the expression of CD68 in APOE^−/−^ mice, which depicts the regulatory effect of OHB in macrophage lineages. AD is pathologically characterized by intracellular neurofibrillary tangles (NFTs) containing phosphorylated tau. We then examined whether ApoE deficiency resulted in the excessive tau accumulation in ApoE^−/−^ mice and found that ApoE deficiency induced increased tau accumulation in the hippocampus of the brain (Figure 1G). We also found that β-OHB ameliorated HFD-induced AT8-positive tau tangles colocalized with thioflavin-S in the hippocampal region of ApoE^−/−^ mice (Figure 1G).

### 3.2. β-OHB Treatment Reduced IgG Extravasation in ApoE^−/−^ Mice

It is well established that increased lipid deposition and Aβ accumulation in the ChP affect CSF permeability [16,17,18]. Thus, we then determined the levels of IgGs in the brain of ApoE^−/−^. A considerably increased intensity of IgG staining was evident in ApoE^−/−^, relative to WT, brain (Figure 2A,B). β-OHB treatment clearly reversed increased extravasation of IgG into brain parenchyma. The overall density of IgG is presented in Figure 2B. ApoE^−/−^ tissue demonstrated a 3.8-fold increase in IgG compared with WT tissue (*p* = 0.003).

### 3.3. β-OHB Inhibited Inflammation in the Brain of ApoE^−/−^ Mice

Based on the recent evidence that identified ApoE4-specific differentially expressed ChP genes [19], we next examined further in ECPC4 cells whether β-OHB regulates the expressions of ApoE4-specific differentially expressed ChP genes, ubiquitin specific peptidase 18 (usp18), IFN-induced protein with tetratricopeptide repeats 3 and 1 (ifit3, ifit1), interferon, alpha-inducible protein 27–like 2A (ifi27l2a), interferon-induced protein 44 (ifi44), guanylate-binding protein 3 (gbp3), IFN-regulatory factor 7 (irf7), and receptor transporter protein 4 (rtp4), the biological functions of which include regulation of autoimmunity by macrophages and dendritic cells, and maintenance of blood-brain barrier integrity [20,21,22]. We observed that, except ifi27l2a, expressions of usp18, ifit3, ifit1, ifi44, gbp3, irf7, and rtp4 were induced by oxLDL treatment, and β-OHB treatment reversed oxLDL-stimulated expressions of usp18, ifit3, ifit1, ifi44, gbp3, irf7, and rtp4 (Figure 3). 

### 3.4. β-OHB Treatment Reduced Atherogenic Plaque Formation in ApoE^−/−^ Mice

ApoE deficient mice are the most widely used murine models of atherosclerosis and the formation of plaques is an important indicator of atherosclerotic pathology. In this study, we primarily investigated the efficacy of OHB in ApoE^−/−^mice by measuring the ORO stained lipid areas within the aortic tissues. As shown in Figure 4A,B, HFD-fed ApoE^−/−^ mice showed increased atherosclerotic plaques (161.5 ± 13%) in aortic regions, while ApoE^−/−^ + β-OHB mice exhibited reduced plaque deposits by ~60% in the aorta compared with those of WT. Foam cell formation, irregular intima thickening, and macrophage infiltration to adventitia are the major phenotypic characteristics of atherosclerosis. We next examined the above-mentioned histological changes in the aortic tissue sections. As shown in Figure 4C, HFD-fed ApoE^−/−^ showed clear atherosclerotic plaque linings along the aortic wall thus narrowing aortic lumen, which was reversed in ApoE^−/−^ + β-OHB. These results suggest the therapeutic effect of β-OHB on the development of lipid deposits and plaque formation. Lipid profiling analysis demonstrated no difference in the levels of total cholesterol, LDL, HDL, and triglyceride between ApoE^−/−^ and ApoE^−/−^ + β-OHB groups (data not shown).

### 3.5. β-OHB Treatment Reduced Serum Resistin Levels in ApoE^−/−^ Mice

We then assessed serum levels of adipokines, leptin and resistin (Figure 5A,B). As expected, leptin, a prototype of adipokine, was decreased in ApoE^-/-,^ and β-OHB treatment did not affect serum leptin levels. However, resistin, a proved risk factor for both AD and atherosclerosis, was increased and β-OHB significantly reduced serum level of resistin in ApoE^−/−^ + β-OHB mice, again confirming the possible therapeutic effect of β-OHB in both AD and atherosclerosis via inhibiting lipid-induced inflammations. 

## 4. Discussion

We have shown in the current study that β-OHB exerts a protective effect, possibly via anti-inflammatory action, on both AD and atherosclerosis in HFD-fed ApoE^−/−^ mice. β-OHB delays the progression of AD via attenuating inflammatory reactions in the ChP region of ApoE^−/−^ mice. β-OHB also impedes plaque formation and lipid deposits in atherosclerotic arteries. Moreover, we found that β-OHB reduced serum level of resistin, a hormone known to be associated with both AD and atherosclerosis.

ApoE functions in lipoprotein metabolism and cholesterol homeostasis [23]. Although ApoE is a risk factor for both AD and atherosclerosis, the common mechanism of action remains to be elucidated. A very recent study, however, identified a complementary regulating function of ApoE that directly links ApoE to the regulation of the immune system [19]. Based on the above referenced report and a known inhibitory function of β-OHB on NACHT-, LRR-, and pyrin (PYD)-domain-containing protein 3 (NLRP3) inflammasome pathway [24], we explored the potential therapeutic effect of β-OHB in murine AD and atherosclerosis model on ApoE^−/−^ mice. Due to the short half-life of β-OHB [25], we delivered β-OHB through osmotic minipumps and adopted the dosage of β-OHB based on the previous report that showed an approximately three fold increase in baseline plasma levels of β-OHB [26]. 

ChP is the region of principal intracranial neuroimmunological interface that forms the blood-CSF barrier, and is the major gateway for bloodborne leukocytes to infiltrate the central nervous system in inflammatory and degenerative brain diseases [27,28,29,30,31]; thus we first examined the ChP for lipid deposition. In line with the previous report [19], we found increased lipid deposition and amyloid plaque formation in the ChP region. Since ChP is the blood-CSF barrier and lipid deposition in ChP might lead to damage and leakage of CSF, we next determined the extravasation of IgG into brain parenchyma and observed decreased expression. We also found that β-OHB abrogated HFD induced lipid deposition as well as amyloid plaque formation in the ChP, which reveals the inhibitory effect of β-OHB on lipid deposition-related diseases of the brain, such as AD. 

Tau is a microtubule associated protein and is post-translationally modified in various ways, among which abnormal hyperphosphorylated tau in NFTs is frequently observed in AD and other taupathies [32,33,34]. In the current study, we observed that the level of phosphorylation of tau protein was increased in HFD-fed ApoE^−/−^ mice compared with NCD-fed C57BL/6J control mice, implicating the involvement of ApoE in the development of AD. We also observed that β-OHB reduced HFD-induced aggregates of tau tangles. 

Lipid deposition promotes inflammation. ApoE4 allele is more prone to lipid-induced inflammation; especially, interferon-related genes are more likely to be increased in ApoE4 allele compared with ApoE2 and ApoE3 alleles [19]. We then explored the possible inhibitory function of β-OHB in ApoE-specific, differentially regulated inflammatory genes in ECPC4 ChP cells, and indeed observed that β-OHB attenuated ApoE-specific, differentially regulated inflammatory genes. This result clearly suggests a therapeutic effect of β-OHB in human AD, since ApoE4 is a well-established high-risk factor for early onset of AD in humans.

Another finding in the current study that supports the therapeutic effect of β-OHB in both AD and atherosclerosis is that β-OHB reduced the serum level of resistin, a hormone that plays important roles in, and is associated with, both AD [35] and atherosclerosis [36]. Evidence in multicohort studies also indicates it as a biomarker as well as a risk factor for AD [37,38,39]. Higher levels of resistin in both serum and cerebrospinal fluid were observed in AD patients. 

Recently, the applications of ketogenic diets as therapeutic modalities has gained substantial attention. Given that ketogenic diet requires drastic changes in dietary pattern, however, adherence to and maintenance of ketogenic diet are challenging. Thus, exogenous ketone supplements in the form of ketone salts or ketone esters have been developed. A study showed that blood β-OHB levels increased from 0.2 to 3.3 mM, the level of ketosis, 1 h after 1.9 kcal/kg of ketone ester consumption [40]. The present findings show that β-OHB reduced pathological phenotypic changes of AD as well as atherosclerosis, which implicates the potential application of exogenous β-OHB supplement as dietary intervention in the treatment of AD and atherosclerosis. 

## 5. Conclusions

Evidence demonstrates the neuroprotective effects of β-OHB in various models of neurological disorders [26,41,42,43,44]. However, underlying mechanistic details of β-OHB effects remain to be elucidated. The present study unravels a novel anti-inflammatory function of β-OHB, leading to new insights into the mechanism by which β-OHB confers protection against AD and atherosclerosis, thereby presenting the possibility of β-OHB as a dietary therapeutic modality. Further studies to validate the efficacy of β-OHB for metabolic diseases, including neurodegenerative and cardiovascular diseases, will provide further scientific understanding and clinical translation.

## Figures and Tables

**Figure 1 nutrients-12-00471-f001:**
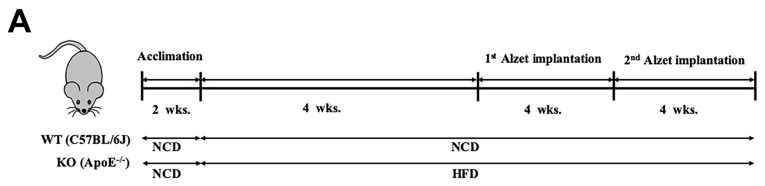
β-hydroxybutyrate (β-OHB) attenuates apolipoprotein-E (ApoE)^−/−^-triggered lipid, amyloid deposition and tau accumulation. (**A**). Schematic illustration of the overall in vivo experimental procedure. (**B**) Mice choroid plexus (ChP) sections were stained with oil red O (ORO) for lipid (red, black arrowhead), deposits and counterstained with hematoxylin (HE). (**C**) Quantitative data representations of the percentage of lipid deposit in ChP regions. (**D**) Brain sections were stained with Congo red for amyloid plaque (black arrowhead) deposits and counterstained with HE. (**E**) Brain sections were stained for CD68^+^ and counterstained with HE. (**F**) Quantitative data representations of the fold increase in CD68^+^ staining. (**G**) Brain sections were double-labeled with thioflavin-S and AT8 (pSer^202^/Thr^205^) antibody for tau entanglements (Scale bars = 20 µm). WT, normal chow diet-fed C57BL/6J; HFD-ApoE^−/−^, high fat diet-fed ApoE knock-out; β-OHB, β-hydroxybutyrate; ChP, choroid plexus; V, ventricle; SNR, substantial nigra pars compacta. ** *p* < 0.001 vs. HFD-ApoE^−/−^.

**Figure 2 nutrients-12-00471-f002:**
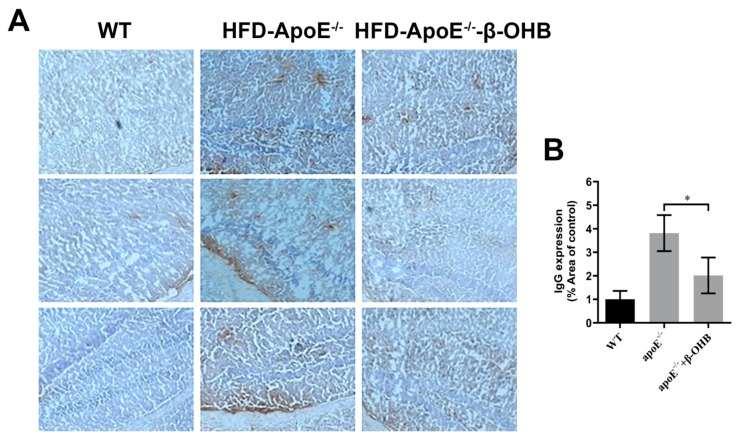
β-OHB decreases IgG extravasation in HFD-ApoE^−/−^ mice. (**A**) Brain sections were stained for mice IgG and counterstained with hematoxylin. (**B**) Quantitative data representations of the percentage of mice IgG staining. WT, normal chow diet-fed C57BL/6J; HFD-ApoE^−/−^, high fat diet-fed ApoE knock-out; β-OHB, β-hydroxybutyrate; ChP, choroid plexus; V, ventricle. * *p* < 0.05 vs. HFD- ApoE^−/−^.

**Figure 3 nutrients-12-00471-f003:**
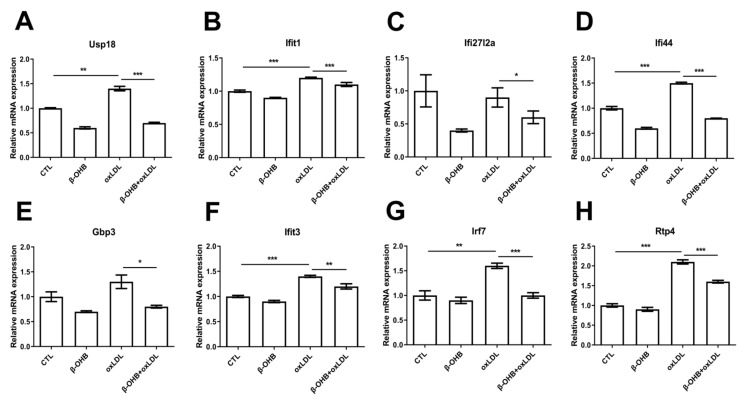
β-OHB decreases ApoE4-specific differentially expressed ChP genes. Quantitative mRNA expressions of (**A**) Ubiquitin specific peptidase 18 (Usp18), (**B**) IFN-induced protein with tetratricopeptide repeats 1 (Ifit1), (**C**) Interferon alpha-inducible protein 27–like 2A (ifi27l2a), (**D**) Interferon-induced protein 44 (ifi44), (**E**) Guanylate-binding protein 3 (gbp3), (**F**) IFN-induced protein with tetratricopeptide repeats 3 (Ifit3), (**G**) IFN-regulatory factor 7 (Irf7), and (**H**) Receptor transporter protein 4 (Rtp4). Data are represented as mean ± S.D. of three independent experiments in duplicates. CTL, control; β-OHB, β-hydroxybutyrate; oxLDL, oxidized low density lipoprotein. * *p* < 0.05. ** *p* < 0.001. *** *p* < 0.0001.

**Figure 4 nutrients-12-00471-f004:**
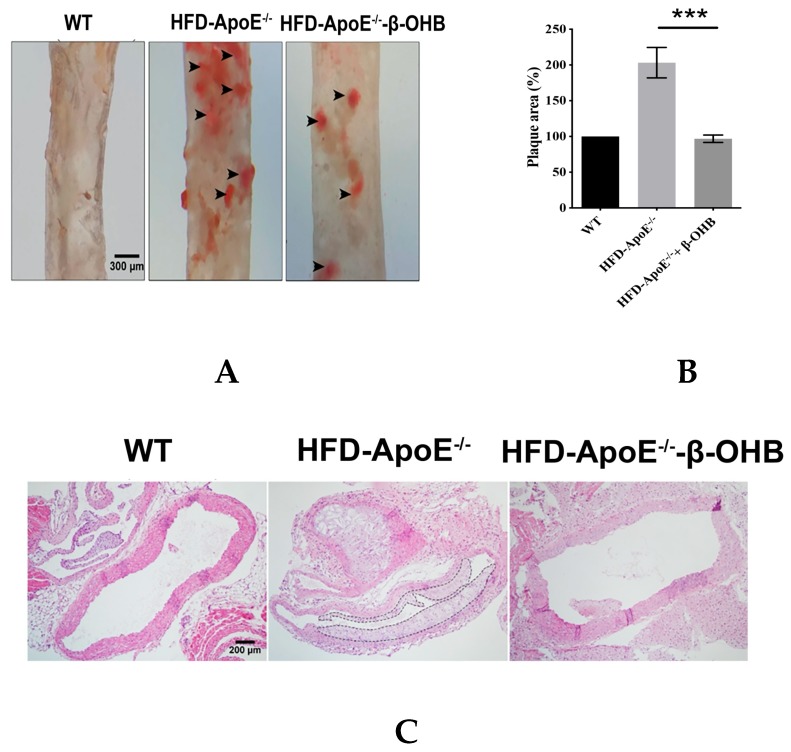
β-OHB attenuates atherosclerotic changes in ApoE^−/−^ mice. (**A**) Oil red O staining of atherosclerotic plaques (arrowheads) in the aorta. Scale bars = 300 µm. (**B**) Quantification of Oil red O stained area. (**C**) H&E staining of aortic roots. Scale bars = 200 µm. (**D**) Serum lipid profiling. WT, normal chow diet-fed C57BL/6J; HFD-ApoE^−/−^, high fat diet-fed ApoE knock-out; β-OHB, β-hydroxybutyrate. *** *p* < 0.001.

**Figure 5 nutrients-12-00471-f005:**
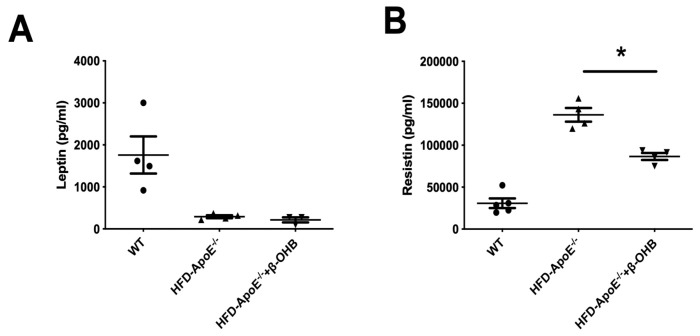
β-OHB decreases the production of resistin. The levels of leptin (**A**) and resistin (**B**) in plasma were analyzed by immunoassay analysis. WT, normal chow diet-fed C57BL/6J; HFD-ApoE^−/−^, high fat diet-fed ApoE knock-out; β-OHB, β-hydroxybutyrate. * *p* < 0.05.

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
