# Peer review of "β-hydroxybutyrate Impedes the Progression of Alzheimer’s Disease and Atherosclerosis in ApoE-Deficient Mice"

_nutrients, 2020, doi:10.3390/nu12020471_

Round 1
Reviewer 1 Report
Revisions have improved paper sufficiently. Ketogenic diet extrapolation interesting still speculative
Author Response
We very much appreciate Reviewer's comment on our revised manuscript.

Reviewer 2 Report
Authors have accepted all my recommendations. However, ketogenic diet security should/may be cited/discussed as last paragraph is somewhat optimistic ...
Author Response
Reviewer 2; Comments and Suggestions for Authors
Authors have accepted all my recommendations. However, ketogenic diet security should/may be cited/discussed as last paragraph is somewhat optimistic ...
⇒ Authors’ response; We very much appreciate reviewer’s comment. As the reviewer suggested, the following statement that address the need for further studies has been added in the conclusion.
“Further studies as to validate the efficacy of β-OHB for metabolic diseases including neurodegenerative and cardiovascular diseases will provide scientific understanding and clinical translation.”

This manuscript is a resubmission of an earlier submission. The following is a list of the peer review reports and author responses from that submission.
Round 1
Reviewer 1 Report
The study offers good morphometrics on the aortic atherosclerotic involvement, but the claim that AD in the brain is retarded requires better documentation of AD in control brains: to include ptau and entanglements. More detail on brain tissue not just in the choroid is needed.
Reviewer 2 Report
2.6. Evaluation of Atherosclerosis and Oil Red O (ORO) Staining - It is not clear that two different assays were carried out. It should be rephrase or separate the text in two paragraphs (according to results shown in 3.4/figure 3).
General comment: I think results and discussion should be directed to conclusions based on nutrition (this journal is focus on that issue). I.e. can be achieved the administered dose only with a balance diet? or How should be that ideal diet? Food composition % of β-hydroxybutyrate.